# A Fifteen-Year Survey for Orthopedic Malpractice Claims in the Criminal Court of Rome

**DOI:** 10.3390/healthcare11070962

**Published:** 2023-03-28

**Authors:** Camilla Bernardinangeli, Carolina Giannace, Simone Cerciello, Vincenzo M. Grassi, Maria Lodise, Giuseppe Vetrugno, Fabio De-Giorgio

**Affiliations:** 1Department of Healthcare Surveillance and Bioethics, Section of Legal Medicine, Università Cattolica del Sacro Cuore, L.go Francesco Vito 1, 00168 Rome, Italy; 2Unit of Orthopedics, Fondazione Policlinico Universitario Agostino Gemelli IRCCS, L.go Agostino Gemelli 8, 00168 Rome, Italy; 3Risk Management Unit, Fondazione Policlinico Universitario Agostino Gemelli IRCCS, L.go Agostino Gemelli 8, 00168 Rome, Italy

**Keywords:** medical liability, malpractice, criminal proceedings, orthopedic trauma practice

## Abstract

The number of legal disputes in the field of medical liability has increased exponentially in the last decades. The aim of this study is to investigate the outcomes of criminal cases against healthcare professionals in Italian criminal courts. The hypothesis is that the majority of cases are dismissed and/or most professionals in these cases are acquitted. This retrospective analysis considers criminal proceedings related to medical professional liability registered with the general register of crime reports of the Public Prosecutor’s Office of Rome in the time interval between 1 January 2000 and 31 December 2015. A total of 4793 criminal proceedings were ultimately identified. Proceedings related to the field of orthopedic trauma were then examined and identified. A complete analysis of 132 of the identified files (76.7%) was carried out. The field with the highest risk of disputes was determined to be the field of trauma. The most frequent complaint was found to arise from unsatisfactory surgical outcomes following elective surgery. The most affected anatomical district is the lower limb in both elective and trauma cases, followed by the upper limb in traumatology and spine cases. The surgeon is the most frequently quoted role of the professional involved. The number of physicians actually convicted (3.93%) and for whom liability was thus recognized, i.e., the existence of a causal link between their conduct and the event that took place was established, appears to be extremely small when compared with the far more significant values related to dismissals (53%) and acquittals (14.2%). Adequate legal reform aiming to reduce this disproportion is necessary to ensure physicians experience a more relaxed daily profession and to restore the original connotations of the doctor–patient relationship with the abolition of defensive medicine.

## 1. Introduction

In the last decades, legal disputes related to medical liability have been rapidly increasing. Simultaneously, a culture of “risk management” has gained importance in the medico-legal field as new tools are sought to address this phenomenon.

This problem is firstly a consequence of continuous improvements in diagnostic and therapeutic techniques which have raised patients’ expectations. These improvements have led patients to overestimate the outcomes of real care compared with what they receive [1].

Secondly, the digitalization of medical records has made the late evaluation of the efficacy and adequacy of a patient’s clinical treatment easier. 

Thirdly, modern society has made patients more conscious of their rights and less inclined to accept the inevitability of a disease, leading to patients being more fearful of or exhibiting a negative approach to physicians. 

As a consequence, the concept of “medical malpractice” was born, which is defined as medical assistance with a negative outcome that differs from what the patient expected [2].

This concept is widespread all over the world [3]. It is crucial to highlight that real medical errors are identifiable in only a small percentage of so-called episodes of “malpractice”. For example, in a study by Brennan et al., it was found that the rate of adverse outcomes in hospitalized patients was 3.7%, and more than 70% of these outcomes were unrelated to medical errors [4]. Another study showed that only 15% of medical malpractice cases actually involved true doctors’ wrongdoing [5]. 

According to a 2005 congressional report, in more than 80% of cases of presumed medical misconduct, a medical error was not identified [6].

In summary, only a small portion of patient harm is truly attributable to medical error; the remainder results from the inherent risks of medical practice. Medicine is not an unerring science; therefore, complications are possible (and not always preventable) in every medical act, but it is unrealistic to blame clinicians for every adverse outcome [7,8,9].

Moreover, it is essential to note that beyond the natural tendency to make errors, which is deeply rooted in every individual, such problems do not only concern single healthcare professionals but whole healthcare systems, which therefore need to be carefully implemented to address patient safety [10].

Other studies conducted in the US have confirmed that orthopedic surgeons suffer from a high malpractice claim rate, with the orthopedic practice being one of the fields least appreciated by patients in terms of communication skills [11] and being the medical field with the fourth highest risk of its practitioners being denounced, with 15% of orthopedists being sued yearly.

With respect to the type of orthopedic procedures involved in this phenomenon, an article published in 2001 by the American Academy of Orthopedic Surgeons showed that patients are more likely to file lawsuits against orthopedic surgeons for femoral fractures, mostly because of problems concerning fixation, early discharge, or the mispositioning of prosthetic hips [12].

Data from Europe mirror these findings: A review from 2009 reports fractures and trauma as the primary orthopedic fields experiencing malpractice lawsuits, of which patient dissatisfaction with surgical treatment was the most important cause [13]. 

Machin et al. [14] confirmed these findings, concluding that orthopedic surgery is second only to gynecology and obstetrics with regard to the number of lawsuits. The lawsuit rate increases every year with a peak in 2010–2011 (16%), with orthopedists specializing in backbone and knee surgery being most commonly involved. 

A significant study of the Italian scenario from the University of Tor Vergata in Rome analyzed 1925 medical malpractice lawsuits filed in the Civil Court of Rome between 2004 and 2010. Of these, 243 (13%) were orthopedic-trauma-related, 631 were general-surgery-related, 199 were gynecology-related, and 195 were plastic-surgery-related. Among the orthopedic cases, compensation was awarded to the plaintiff for 182 patients (75%), while this was not the case for 61 patients (25%). In addition, in 70% of the cases the physician was sued (94% of whom were orthopedic surgeons), in 2.4% of the cases the complaint was against the anesthesiologist, and in 2% the complaint was against the general surgeon. The complaint was made regarding an elective operation in 149 cases (61%) while it was related to trauma surgery in 94 cases (39%). Lower limbs were involved in 136 cases (56%), 75 cases involved the upper limbs (31%), and 32 cases involved the spine (13%) [15].

These results were substantially confirmed in another study published by the same group of researchers in 2021 in a retrospective review which analyzed all the judgments drawn up by the judges of the Civil Court of Rome, XIII Chamber (competent and specialized section for professional liability trials) between January 2018 and February 2019. 

The authors highlighted the burden of malpractice in Italy, noting that in 84.6% of the judgments, one or more health facilities were sued, while in 58.2% of cases, one or more health workers were present among the defendants. Regarding the type of damages claimed by patients, data suggest that the majority are related to physical injuries, while compensation was requested by heirs for patients’ death in only approximately 20% of cases [16]. 

In the criminal field, a retrospective study conducted in Milan in 2014 was published in Forensic Science International on cases of suspected medical malpractice that occurred between 1996 and 2009 that resulted in death followed by a judicial autopsy. In terms of subcategories, the surgical field appeared to be most involved (representing a total of 33.7% of cases). Specifically, regarding the branches involved, there was a clear dominance of abdominal surgery (30.5%), orthopedics (14.1%), neurosurgery (13.3%), and gynecology–obstetrics (10.9%). Of the 71 cases whose judicial processes were completed, judicial autopsy identified the cause of death in 69% (49) of the cases; medical malpractice profiles were identified in 17% of the cases (12, half of which resulted from surgical errors), while a causal link between malpractice and death was identified in only 12.7% of the cases (nine cases, five of which resulted from surgical errors) [17].

Among surgery subcategories, plastic surgery is often involved in litigations, highlighting the dominance of physical malformation and esthetic damage as one of the leading causes of patient litigation [18].

The operatory room represents one of the most controversial healthcare facilities, as litigations involving surgical operations frequently involve not only surgeons but also anesthesiologists [19]. Moreover, it is notable that this phenomenon does not only involve medical doctors but also dental healthcare professionals, whose sector experiences similar trends [20].

In 2008, Italy was the European country with the most physician malpractice prosecutions. This can be partly explained by the fact that although physicians are criminally prosecuted in many European countries, most lawsuits are commenced in Italy because the costs related to evidence gathering and prosecution are borne by the state and not by those bringing the case. Civil trials generally have longer timelines, and criminal proceedings do not exclude the possibility of filing a civil suit in parallel. Moreover, in Italy, if criminal proceedings are successful for the offended party, it is possible to obtain monetary compensation [21].

According to a study conducted in 2004 by the Italian Ministry of Health’s Technical Commission on Clinical Risk, in the ranking of specialty fields charged with the most errors, orthopedics topped the list with 16.5% of errors. The most frequent errors were made in the operating room (32%), followed by inpatient wards (28%), emergency departments (22%), and outpatient clinics (18%).

The aim of this study is to retrospectively evaluate the conclusions of the judicial processes regarding complaints of alleged malpractice filed in the city of Rome between 2000 and 2015. The orthopedic and trauma fields are independently assessed as are the motivations underlying the complaints, the anatomical districts involved, and the subjects of the convictions.

The hypothesis of this study is that in the criminal courts, most criminal cases against healthcare professionals are dismissed, and/or most professionals in these cases are acquitted.

## 2. Materials and Methods

Using the scientific article search engine “Pubmed”, a study of the literature in the field of orthopedic professional liability was performed. 

The following keywords were used: orthopedic medical errors, orthopedic medical liability, orthopedic liability, orthopedic malpractice, orthopedic claims, and orthopedic litigations. In addition, by using the “similar articles” function of the search engine and evaluating the references of the analyzed articles, additional scientific articles were identified. Publications from 2000 to 2016 were included.

This study was built on a retrospective analysis previously conducted at the Institute of Forensic Medicine of the Catholic University of the Sacred Heart in Rome, which included criminal proceedings related to medical professional liability filed with the Public Prosecutor’s Office of Rome during the time interval between 1 January 2000 and 31 December 2010. This previous work was integrated with the retrospective examination using the same evaluation criteria of all the criminal proceedings related to medical professional liability filed with the General Register of Offense Reports of the Public Prosecutor’s Office of Rome during the period from 1 January 2010 to 31 December 2015. A total of 4793 criminal proceedings were ultimately identified.

Proceedings related to the field of orthopedic trauma were then examined and identified, and 172 criminal cases (3.58%) of this nature were thus identified. 

It was not possible to analyze all 172 files since 40 files (23.3%), including the oldest files, had been destined for pulping, meaning that their paper version was no longer available. 

Thus, a complete analysis of 132 files (76.7%) was carried out. Some of the files were in paper format, while others were assessed in digital format via the TIAP (Computer Processing of Criminal Procedure Acts) system. Using data inferred from the ReGe computer system (General Register of Procedure Acts), it was possible to calculate the following parameters: the trend over time in the number of medical professional liability prosecutions relating to the orthopedic trauma field in Rome from 2000 to 2015; the types of offenses prosecuted; the number and category of professionals involved; the number of professionals for whom dismissal was requested; the number of professionals for whom there was opposition to the motion to dismiss; the number of professionals for whom the request for dismissal was granted; the number of professionals for whom an indictment or direct summons to trial was requested (and granted); the number of proceedings settled with a conviction; and the number of proceedings settled with a judgment of acquittal. In addition, the following features of the proceedings were analyzed: the number of proceedings for which a court-appointed expert witness was appointed and the qualification of the consultants; the number of proceedings for which court-appointed technical consultants identified culpable conduct in causal connection with the event; the outcome of proceedings for which culpable conduct was identified; the outcome of proceedings for which culpable conduct was not identified; the number of criminal proceedings for which clarifications and/or additions were requested from court-appointed technical consultants; the outcome of clarifications and/or supplements; the number of proceedings for which a party consultant was appointed and the qualifications of the consultants; the number of proceedings involving orthopedics and those involving traumatology, as well as the respective subjects and anatomical districts involved in both cases; the type of healthcare facility involved; the role played by the professional; the identification, where death occurred, of a cause of death; and the number of proceedings directed against medical professionals and those in which the injury resulted from accidents/self-inflicted injuries or other matters for which physicians were later held liable.

The computational software used to process the statistics and graphs presented below was Microsoft Excel as well as SPSS version 22.

## 3. Results

The graph in Figure 1 shows the trend over time of medical malpractice complaints in the orthopedic field at the Public Prosecutor’s Office of Rome.

### 3.1. Trend in Years

The collected data showed a gradual increase in orthopedic prosecutions in the 2000–2001 interval (14 and 13, respectively) (Figure 2); there was then a slight reduction in 2002 (7), followed by a new surge in 2003–2004 (17 and 24, respectively) and a further reduction in 2005–2006 (11, 16). This was followed by fluctuating but still decreasing values relative to the previous values (8, 10, 6, 7), which were followed by a slight increase in 2011 (12) and then another decrease arising particularly in the last year under analysis (2).

### 3.2. Types of Crimes

The analysis of all the prosecutions in the orthopedic field showed that the most prosecuted crimes were those of Article 590 of the Italian Criminal Code (73.2%), i.e., culpable personal injury, and Article 589 (19.76%), i.e., manslaughter (Figure 3).

The other prosecuted crimes were those of Article 582, or personal lesions, and Article 583–585 (aggravating circumstances of the previous articles); Article 485, or forgery of private writing; Article 328, or refusal to perform official acts; Article 317, or extortion; Article 476, or material falsity in public acts committed by a public officer; Article 593, or “hit-and-run”; and Article 348, or abuse of the profession.

### 3.3. Involved Medical Specialty Fields 

The total number of physicians and healthcare professionals involved in orthopedic professional liability prosecutions in the city of Rome from 2000 to 2015 was 456 (Figure 4). Data analysis made it clear that the most commonly prosecuted professionals were orthopedists (292) followed by anesthesiologists (28); radiologists (22); neurosurgeons (10); general surgical nurses and internal medicine physicians (9 in each of the categories); cardiologists (7); and multidisciplinary professionals (18). There were also a number of cases for which the relevant specialty field could not be identified (24), while 28 professionals belonged to various less-involved branches (pulmonology, urology, neurology, infectious diseases, hematology, gynecology, emergency medicine, and surgery, etc.). 

### 3.4. Trial Characteristics

The study of all the professional liability proceedings in the orthopedic field showed that, in most cases, a dismissal of the proceedings was requested. 

In fact, out of a total of 456 cases considered, dismissal was requested for 254 of the prosecuted professionals (55.7%).

However, it is important to highlight that for 86 (18.8%) of the investigated professionals, no data were available regarding the course of the investigation. 

Among these requests, dismissal was granted in 242 cases, of which 145 (59.9%) were granted without opposition and 54 (22.3%) were granted despite the fact that there was opposition. 

Motions to dismiss were not granted for 12 (4.9%) of the 254 suspects, 5 of whom filed in opposition to the motion to dismiss, 3 of whom did not file in opposition, and for 4 of whom the relevant data were not available. 

The judicial processes of these 12 cases included 5 direct summonses to trial (wherein only 1 of the cases had filed in opposition to the motion to dismiss), while for the remaining 7 cases the investigations were still ongoing.

For 61 (13.3%) of the 456 professionals involved, the direct arraignment was ordered. Of these, 42 (68.8%) professionals were acquitted; for 12 (19.6%), the investigations were still ongoing; 5 (8.4%) of the professionals were convicted; and 2 cases (3.3%) were declared inadmissible. 

The reasons for the 42 acquittals were as follows: 17 (40%), dismissal of the complaint; 14 (33.3%), no case to answer; 5 (11.9%), acquitted for not having committed the act; 3 (7.1%), due to the statute of limitations; 1 (2.3%), no offense to answer; and 1 (2.3%), non-prosecution. In an additional case (2.3%), the reason for the acquittal was not available. 

For 55 of the 456 suspects (12%), the indictment was requested, which in 17 cases (30.1%) resulted in a verdict of non-prosecution for not having committed the act.

The indictment was granted in 38 (69%) of the cases. Among them, 23 cases (60%) received a judgment of acquittal; 13 (34.2%) received a judgment of conviction; and in 2 cases (5.2%), the investigation was still ongoing.

The reasons for the 23 acquittals were as follows: 13 (56.5%), no case to answer; 6 (26%), no offense to answer; and 4 (17.4%), acquitted for not having committed the act.

For 18 (3.9%) of the 456 professionals involved, a conviction was issued. 

Specifically, in relating this numerical value (18) to the number of professionals for whom the judicial process continued (116), a conviction rate of 15.5% emerged.

The sentences imposed ranged from the payment of a fine and the court costs to imprisonment (min. 2 months, max. 1 year and 2 months). In 3 of the 18 cases (16.6%), compensation for damages to the civil party was also ordered (Figure 5).

For 65 (14.2%) of the total 456 professionals, an acquittal verdict was issued. 

Specifically, in relating this numerical value (65) to the number of professionals for whom the judicial process continued (116), a 56% acquittal rate emerged. The reasons for these acquittals were as follows: 24 (36%), no case to answer; 18 (27.6%), dismissal of the complaint; 10 (15.3%), the act did not constitute a crime; 10 (15.3%), not having committed the act; 4 (6.1%), non-prosecution; 4 (6.1%), the statute of limitations; and 2 (1.72%), defect of the complaint. In 1 case (0.86%), the reason for acquittal could not be traced.

For 117 (88.6%) of the 132 criminal cases that could be fully analyzed, a court-appointed expert witness was ordered.

Among these, in 22 of the cases (18.8%), only one consultant was appointed, who, specifically, was represented by a forensic physician in 14 cases (11.9%), an orthopedic surgeon in 6 cases (5.12%), and an occupational physician in only one case (0.85%). In one case, the qualification pertaining to the forensic report for the public prosecutor could not be traced. 

A medical committee consisting of a forensic physician and an orthopedist was appointed in 58 of the cases (49.57%), while in the remaining 37 cases (31.6%), medical doctors who specialized in neither forensic medicine nor orthopedics were appointed, i.e., a committee of consultants other than a forensic physician–orthopedists was appointed.

Culpable conduct was identified in 63 (53.8%) of the 117 criminal proceedings in which one or more Public Persecutor’s Consultants were appointed; in 50 cases (42.7%), no culpable conduct was identified; in 4 cases (3.4%), data were not available because the consultancy was not attached to the file.

For the 50 proceedings in which culpable conduct was not identified, the outcome in 28 of the cases (56%) was dismissal of the proceedings; in 12 cases (24%) was a direct summons to trial; in 9 cases (18%) was indictment; and in 1 case (2%) the data were not available.

Among the 12 professionals who went directly to arraignment, 9 (75%) were acquitted (4 for non-existent act, 3 for dismissal of complaint, and 1 for not having committed the act), 2 were convicted, and 1 case was declared inadmissible due to lack of complaint.

Among the nine professionals remanded for trial, two were convicted, while seven were acquitted (three for not having committed the act, two for the act not constituting a crime, and two for the act not existing).

Counterfactual reasoning was addressed by the Public Persecutor’s Consultant in 37 (58.7%) of the 63 cases in which culpable conduct was identified; in the remaining 26 (41.2%) of the 63 cases in which culpable conduct was identified, the Public Persecutor’s Consultant identified exigent conduct but did not invoke counterfactual reasoning.

Of the 63 cases in which counterfactual reasoning was discussed, the outcome was dismissal in 24 cases (38%), direct summons to trial in 7 cases (11%), and indictment in 5 cases (7.9%) (of which 3 cases resulted in a conviction, 1 case resulted in acquittal due to there being no facts, and another case resulted in a judgment of non-prosecution). 

Among the seven cases of direct summonses, five went to an acquittal (three due to dismissal of complaint, one due to statute of limitations, and one due to non-existent act); one went to conviction; and in another case, the investigation was still ongoing.

Of the cases in which counterfactual reasoning was not carried out (26), 16 (61%) were dismissed, 2 (7.69%) were remanded for trial and subsequently acquitted for non-prosecution, while 8 (30.7%) went directly to arraignment. 

Among these cases, in six, a judgment of acquittal was pronounced (three due to dismissal of the complaint, one due to lapse of the statute of limitations, one because there was no case to answer, and in one case the reason for the acquittal was not available); in one case, a judgment of conviction was pronounced; and in another case, the investigation was still ongoing.

Clarifications were requested from the consultants in 26.5% (31) of the criminal cases out of 117 for which consultants were ordered. In 29 of the 31 cases (93.5%), the consultants confirmed the conclusions they had reached in their paper. Only in two cases (6.45%) were the previously stated conclusions disproved upon the acquisition of additional documentation.

In 62 (46.9%) proceedings of the 132 fully analyzed, the parties appointed their own technical advisor.

In 36 cases (58%), the consultant was a medical examiner; in 23 cases (37%), the consultant was an orthopedic surgeon; and in the remaining three cases (4.8%), the consultant was a plastic surgeon; a committee consisting of a medical examiner and anesthesiologist; and a medical examiner and oncologist, respectively.

In 43 cases (69.3%), the consultant appointments were made by the offended party, while in 19 (30.6%) cases, they were made by the suspect.

### 3.5. Orthopedic Sub-Fields Involved

Most of the case history was related to trauma (87 cases), which was more than double the number attributable to elective care (40 cases) (Figure 6). For the 40 cases sent to the pulp mill, it was not possible to define the scope, while 5 cases were of neither scope.

In the orthopedic domain, the most frequently represented events were as follows: 21 (52.5%), dissatisfaction with the surgical outcome; 4 (10%), postoperative complications from elective surgery, 4 (10%), diagnostic errors; 4 (10%), infectious complications; 3 (7.5%), both diagnostic and therapeutic errors; 2 (5%), intra-operative complications; 1 (2.5%), error related to poor communication between healthcare providers; 1 (2.5%), forgetting gauze; and 1 (2.5%), a traumatic event (fracture) occurred during hospitalization for nontraumatic pathology. 

The most affected anatomical districts in the orthopedic field were as follows: the lower limb in 27 (67.5%) cases; the spine in 8 (20%) cases; the upper limb in 4 (10%) cases; and in 1 case (2.5%), the district of interest could not be identified. 

Specifically, regarding the lower limb, the most frequently litigated procedures were prosthetic hip surgery (12 cases), hallux valgus correction (5 cases), and prosthetic/arthroscopic knee surgery (3 cases each).

With regard to the trauma domain, the most frequently represented events were as follows: 31 (35.6%), diagnostic omissions/errors; 21 (24.1%), therapeutic errors; 13 (15%), surgical error; 9 (10.3%), infectious complications; 3 (3.4%), intraoperative complications; 5 (5.7%), cases of both diagnostic and therapeutic error; 1 (1.14%), case of defective patient information; and 4 events could not be framed due to incomplete files (4.6%). 

The districts most involved in trauma were found to be as follows: the lower limb in 50 cases (57.47%); the upper limb in 25 cases (28.7%); the spine in 6 cases (6.9%); polytrauma in 5 cases (5.74%); and the pelvis in 3 cases (3.44%).

The 18 convictions (3.93% of the 457 cases investigated) were almost split equally between the trauma (4 cases, 22.2%) and orthopedic (5 cases, 27.7%) fields.

For the former field, the themes of the convictions (four cases) were intraoperative complications (nerve injury during cruel reduction of fracture), incorrect plaster appliance packing, inadequate postoperative management, and therapeutic omissions (failure to transfuse following hip fracture). 

For the orthopedic field (five cases), however, the themes of the convictions were intraoperative complications (vascular injury during hip prosthesis revision and nerve injury during discectomy), two wrongly performed surgeries, and one therapeutic error.

Among the nine remaining convictions in the cases under study, four did not involve orthopedic professionals (anesthesiologist, internal medicine physician, general surgeons); in three of the cases in which a conviction was handed down, the file was not available in full and therefore it was not possible to determine the reasons for the conviction; and in two cases, the conviction instead concerned a failure to carry out in-depth diagnostics that could be attributed neither to trauma nor to elective care.

### 3.6. Type of Healthcare Facility

In the majority of cases (83–62.8%), the facility involved was a public hospital; in 50 cases (37.87%), a private/contracted care home was involved; in only 1 case (0.75%), a private practice was involved; and in only 1 additional case (0.75%), both private and public facilities were involved.

### 3.7. Role of Orthopedist

Of the 292 orthopedic professionals investigated, their roles could be traced in 173 cases. Of these, 62 (36%) had conducted surgery as a first practitioner and 30 (17%) as a second/third practitioner; 45 (26%) had assisted the patient as consultants in the emergency room; 18 (10%) had been ward physicians (6 were ward chiefs); 15 (9%) were outpatient physicians at the facility; 1 was a private practice physician; 1 was a medical director; and 1 was a resident (2%) (Figure 7).

### 3.8. Causes of Deaths

Of the 28 cases in which death occurred, the cause of death was identified in 26. In only two cases was no diagnostic finding/judicial autopsy performed. In 15 cases, death was due to hemorrhagic/hypovolemic shock or cardiac problems (recent IMA or decompensation); in 9 cases, death was due to respiratory failure (due to pulmonary embolism or infectious processes); in only 1 case, death was caused by renal failure due to rhabdomyolysis; and in 1 case, death was due to a cachectic state in a multimetastatic patient.

### 3.9. Causes of Complaints

The analysis of the 132 complete files available showed that complaints filed against physicians for alleged malpractice in the absence of other offenses that had a causal role in the injury numbered 114 (86.3%), accounting for almost all the cases. 

In the remaining portion, the litigations were related to traffic accidents in 15 cases (12.8%), attempted suicides in 2 cases (1.5%), and as a result of mugging in 1 case (0.75%); in these cases, medical treatment was not deemed appropriate due to the reported injuries.

## 4. Discussion

The topic of medical liability in the orthopedic field invites reflection on the current phenomenon of increasing cases of alleged malpractice, the orthopedic traumatology field being among those most frequently subjected to complaints/claims for compensation. Such complaints are still occurring despite the availability of automated techniques based on data-driven approaches to improve patient safety and the quality of care [22,23,24,25]. 

It is important to note that in Italy, with respect to criminal law, the entire cost of the legal process is paid by the state, whereas the defendant (the doctor) must pay for legal defense. 

This differs from the civil law process, where the damaged patient, seen as a weak subject, receives compensation not only from a single medical worker but from the entire healthcare facility, which is financially and technologically better able to sustain the cost and prevent the risk of a negative outcome [26].

From the analysis of data obtained from the study of criminal cases of suspected medical liability in the orthopedic traumatology field, the following information could be deduced. 

In 2000–2001, there was an increase in cases reporting alleged medical malpractice. In 2002, there was a slight decline in cases, followed by a peak in values reached in the subsequent two-year period and a minimal decline in the years 2005–2006. The subsequent period saw a slight but steady decrease in cases, with only one new rise in 2011 and a major decrease in the last year under analysis. 

It was determined that the most prosecuted crimes were those of Article 590 and Article 589 of the Criminal Code.

The numerical figure representing the physicians actually convicted (3.93%) and for whom liability was thus recognized, i.e., the existence of a causal link between their conduct and the event that took place was established, appears to be of great significance, its small size probably being attributable to the criterion for ascertaining a causal link in criminal law which, as reiterated by the Supreme Court of Cassation, requires it to be ascertained with a “high degree of rational credibility” or “logical probability”, “beyond a reasonable doubt” [27].

One of the major limitations of all Italian studies in the domain of malpractice is that there is no official and comprehensive monitoring and data collection system in Italy that may clearly frame the extent and characteristics of the phenomenon. 

A phenomenon that, as already noted, shows common trends as indicated by the term “malpractice crisis”, it is in fact growing both in the US and in Europe, and this is further accompanied by an increase in the amount of compensation awarded to plaintiffs in the civil sphere [28].

Such a worldwide situation is helpful in first understanding how litigation analysis is necessary for developing prevention strategies and thus reducing the number of claims. Its worldwide scope also highlights how although healthcare systems differ from country to country, the underlying issues are largely analogous, and therefore the interpretation of these critical issues could help all healthcare professionals provide a qualitatively better level of care to patients. 

This phenomenon is of great concern in both Europe and the US for two main reasons. First, it is concerning because it leads to an increase in the cost of insurance policies, as both professionals and healthcare facilities take out contracts with increasingly higher premiums in order to shelter themselves from claims/complaints. Second, it is equally important to note its negative effects on all those involved. There is first the rupture of the doctor/patient relationship of trust generated by the bureaucratization of the clinical approach that has replaced the essential qualities of the traditional and highly personal role of the physician. This rupture further derives not only and not so much from patients having been the victim of an adverse event but from the subsequent difficulties of obtaining an adequate interlocution with those “responsible” for the damage, which prevents the “prevention” or “neutralization” of the conflict and often makes it inevitable that the patient resort to the judicial authority. Moreover, physicians often rightly feel themselves under pressure and therefore tend to increase their use of diagnostic investigations and therapeutic interventions, causing the very costly phenomenon for the state, and therefore for the entire community, of so-called “defensive medicine”, the use of which is believed to generate an expenditure of EUR 13 billion a year [29].

Such a practice, comprising overdiagnosis and overtreatment, is first and foremost the cause of a large quantity of waste, which has the regrettable implication of in the long run generating a lower quality of healthcare itself due to an ever-increasing reduction in available funds linked to increased public spending [30,31].

Human nature cannot separate itself from errors, which are ubiquitous: it is essential, then, to improve error management in order to make it effective outside of and in the prevention of court proceedings. 

In March 2015, data were presented from a recent 2014 Agenas (“Agenzia Nazionale per i Sevizi Sanitari Regionali”, National Agency for Regional Healthcare) survey of 1500 hospital physicians in which it was highlighted that 58% of white coats practice defensive medicine and that for 93% of these physicians, this is likely to increase. The study also explained why defensive medicine is practiced: for 31% of respondents, it is the fault of unfavorable legislation; for 28% of respondents, it is due to the risk of being sued; and for 14% of respondents, it is due to the imbalance of the doctor–patient relationship resulting in excessive demands, pressures, and expectations from the patient and family members. According to the respondents, the potentially effective solutions for reducing the phenomenon are to stick to scientific evidence (49%) and to reform the rules governing professional liability (47%) [32].

## 5. Conclusions

The number of prosecutions in the orthopedic traumatology field increased during the years of observation of this study. The field with the highest risk of legal disputes was the field of trauma in relation to diagnostic omissions/errors and/or inappropriate therapeutic choices. 

In the orthopedic field, the most frequent cause of complaint was an unsatisfactory surgical outcome following elective surgeries (especially hip replacement, the correction of hallux valgus, and prosthetic/arthroscopic knee surgery, as also evidenced by the referenced literature at the international level). Postoperative complications and infectious events were less frequently noted. 

The anatomical district most affected in both fields and in accordance with the literature was found to be the lower limb, followed by the upper limb in traumatology and the spine in orthopedics. 

Regarding the roles of the professionals in providing patient care, it was found that in the majority of cases, they had conducted surgery as the primary caregiver, less frequently as the second/third caregiver, and even less frequently had intervened as consultants in the emergency room.

Upon comparing the number of actual convictions with the far more significant values related to dismissals (53%) and acquittals (14.2%), it is highly hoped that new reform with respect to criminal liability in the field of medical professional activity will produce the necessary results, that is, significantly streamline the disproportionate number of medical professional liability prosecutions, resulting in work relief for public persecutors and judges. With the successful control of the “defensive medicine” phenomenon and consequent reduction in public health spending, medical professionals would also be able to regain satisfaction in the practice of their profession, and perhaps the original connotations of the doctor–patient relationship would be restored.

## Figures and Tables

**Figure 1 healthcare-11-00962-f001:**
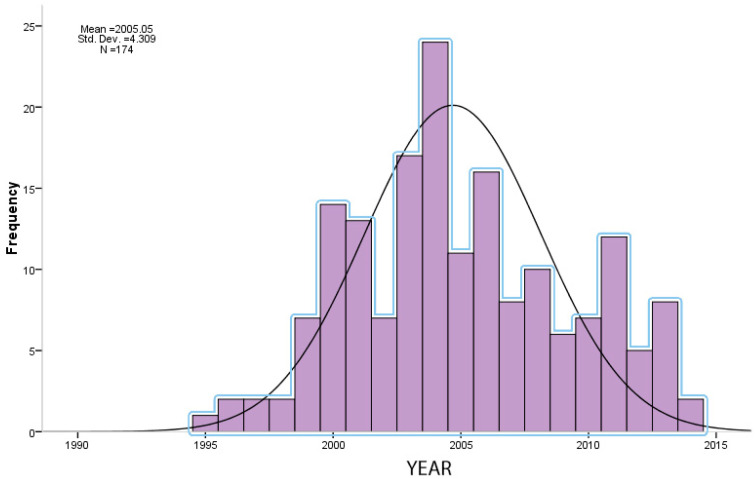
Graph showing the trend over time of medical malpractice complaints in the orthopedic field at the Public Prosecutor’s Office of Rome.

**Figure 2 healthcare-11-00962-f002:**
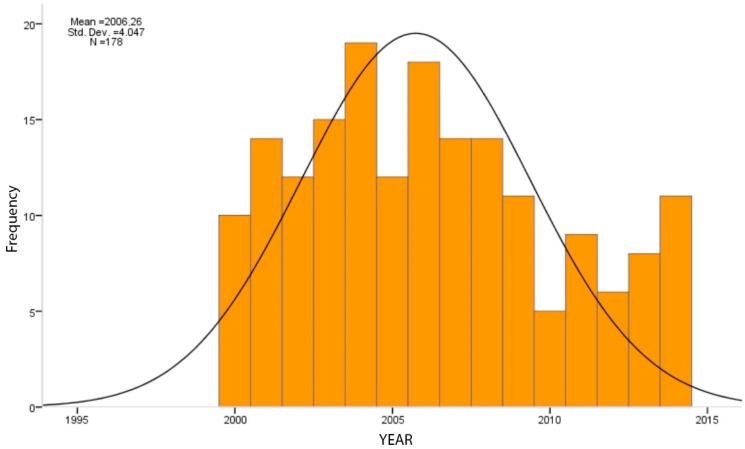
Graph showing the trend over time of medical malpractice crime registrations in the orthopedic field at the Public Prosecutor’s Office of Rome.

**Figure 3 healthcare-11-00962-f003:**
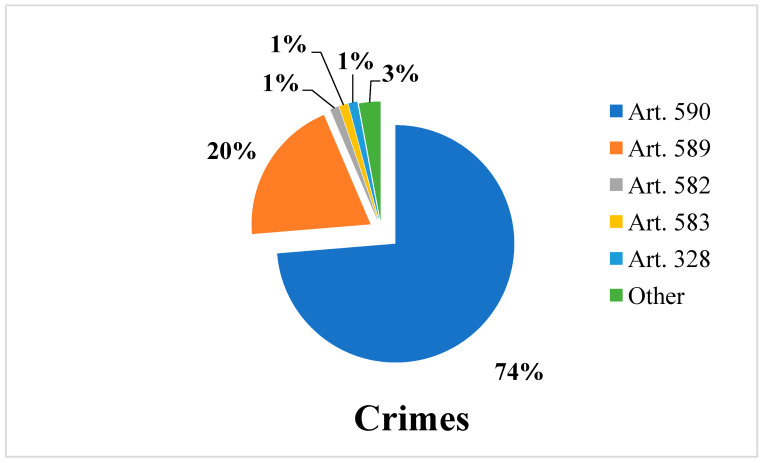
Graph showing the types of crimes prosecuted in the proceedings under study.

**Figure 4 healthcare-11-00962-f004:**
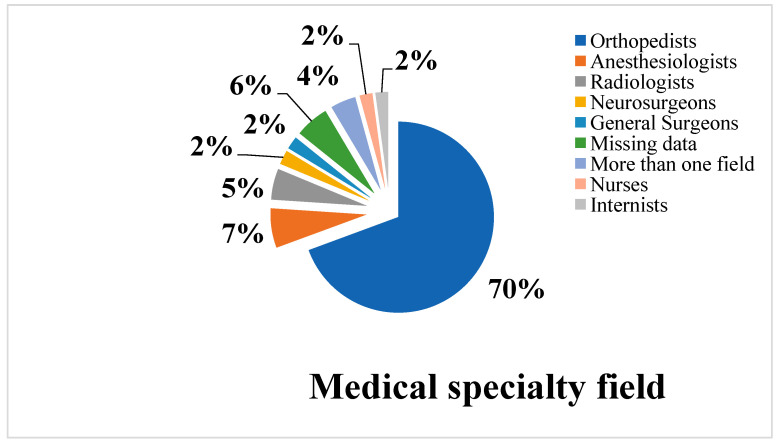
Graph showing the number and specializations of professionals involved in the criminal cases under study (total: 456).

**Figure 5 healthcare-11-00962-f005:**
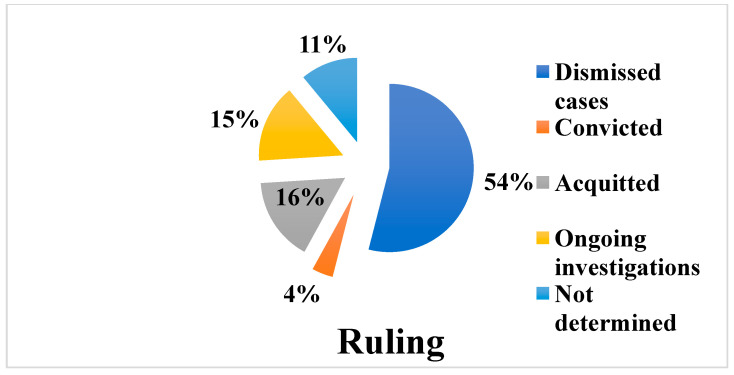
Graph showing rulings on proceedings.

**Figure 6 healthcare-11-00962-f006:**
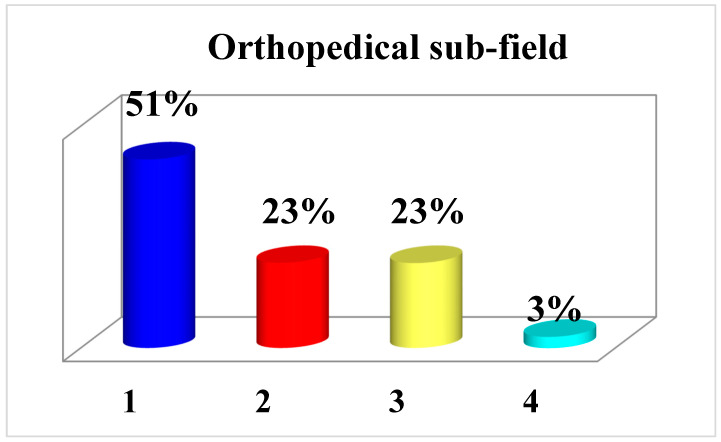
Graph showing the fields involved (orthopedics/traumatology) in the procedures under study. (1) Trauma; (2) election; (3) undetermined; (4) others.

**Figure 7 healthcare-11-00962-f007:**
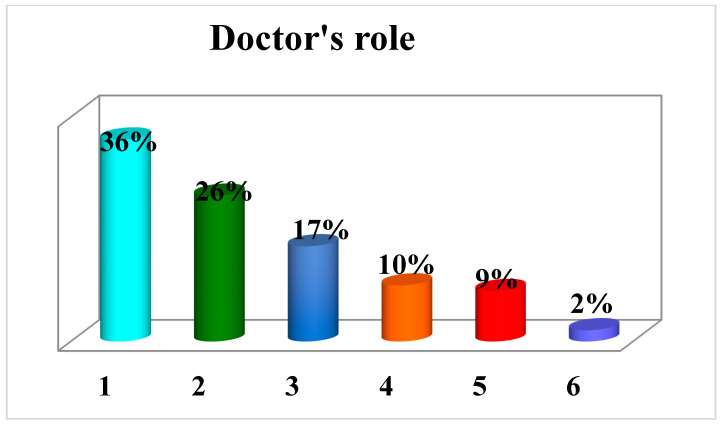
Graph showing the roles of professionals in providing care in the cases under study. (1) Surgeon; (2) emergency room; (3) assistants; (4) ward; (5) clinic; (6) others.

## Data Availability

Data are available upon reasonable request.

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
