# Peer review of "A Fifteen-Year Survey for Orthopedic Malpractice Claims in the Criminal Court of Rome"

_healthcare, 2023, doi:10.3390/healthcare11070962_

Round 1

Reviewer 1 Report

interesting article that deals with an interesting and always topical theme in forensic medicine, or the litigation related to orthopedic and traumatological cases. needs some small corrections on English, but the overall impression is positive

Author Response

Thanking our first Reviewer for this kind comment, We would like to emphasize the relevance of the theme discussed in our Study considering that Orthopedics and Traumatology represent one of the branches of medicine most affected by criminal and civil trials.
We apologize for the mistakes in English grammar and vocabulary, being able to attribute some of it to the difficulty of translating detailed notions of the Italian legal landscape into a foreign language.
We have done the appropriate corrections according to the comments.

Reviewer 2 Report

Data used are very old and limited to only one Italian city (Rome).

The number of cases taken into consideration are rather small to allow any inference.

The article focuses too much on a detailed data analysis which does not seem very fruitful.

Legend within the graphs are in Italian.

Conclusions make sense but they are not really supported by the rather small amount of data collected.

Author Response

We thank this second Reviewer for their interesting point of view.
We are well aware of the limitations our Study especially regarding the number of cases analyzed, which is relatively small and limited to the city of Rome.
On this subject, as observed by the Academic Editor, the Court of Rome is the biggest one in Italy and therefore can be considered as representative of the of the whole national scenario. We hope it can act as a useful tool for comparison with further studies from other Countries.
Regarding the fact that data are slightly old, it is important to highlight that judicial timelines are sometimes extremely slow, particularly in Civil trials. The delay prevents the availability of more recent cases (since these are mostly still ongoing). Considering this aspect, the considered years (2000-2015) seem not be really dated. Moreover, the Study covers a time range of 15 years, which is extremely long representing a useful picture of how the situation has evolved over the past two decades.
Regarding the comment on
fruitfulness of the analysis: we do consider the possibility to compare the reported data with those other studies and other Nations’ scenario as the most relevant aspect of this Work.
Anyway, the Discussion section has been revised with new considerations and references, to enrich it with other insights according to the comment of the reviewer.
Regarding the legend of the two graphs they have been corrected.

Reviewer 3 Report

Dear Authors,

I have read with interest the manuscript ‘A Fifteen-Year Survey for Orthopedics Malpractice Claims in the Criminal Court of Rome’.

The topic is relevant to the present time because it investigates an increasing trend in Medico-legal activity.

However, there are some issues that need to be solved before the manuscript could be considered suitable for publication.

Point 1: the English form, grammar and style require an editing, since it reflects the fact that it was not written by a native English-speaker and sometimes is hardly understandable for non-Italian reader.
Maybe the authors should consider to use a professional editing service.

In any case, I can give some examples that need a compulsory revision:

-          Avoid the use of the term “Medical responsibility” and use instead “liability”.[e.g, line 14]

-          Avoid the use of the terms “first/second/third surgeon” since the English correct form is “surgeon” for the one who performs the operation and “first/second assistant” for the ones who assist the surgeon [e.g., line 26]

-          Avoid expressions like “Civil Right” [line 56] and replace with “Civil Law”

-          Avoid expressions like “surgical area” [line 108] and replace with the term “field”

-          Avoid expressions like “The Rome Public Prosecutor’s Office ” [e.g., line19] and use instead “The Public Prosecutor’s office of Rome”.

-          Avoid the use of “media format” [line 158] and use instead “digital format”.

-          The expression “medical specialty” [lines 213 and 220] or “specialist” [line 375] are not understandable, maybe the authors meant “medical field”or “ortopedist”

-          Avoid expressions like “fact does not exist” and replace with the correct indirect form;

-          Prefer expressions like “medical committee” rather than “medical panel” (line 277)

-          Avoid expressions like “lesivity” [line 396] and use “lesions or injuries”

There are also some not understandable acronyms or expressions that must be clarified:

-          Line 159: what does REGE stands for?

-          Line 276: what does CTPM stands for? find the correct english term for it

-          Figure 3: the legend should integrate what the single Article stands for

-          Figure 4: The legend must be translated in English!

-          Figure 5: The legend must be translated in English!

-          Figure 6: what does ”N.D.” stands for?

-          Figure 7: what does Ambulatory stands for?

-          Line 339: what does “never event!!!” means?

-          Line 480: what are “United Sections”? Did you mean the Supreme Court?

Point 2: The introduction has several issues that need to be addressed.

The introduction should not include personal opinions of the Authors.
Lines 50-52: a proper reference is missing and moreover the expression “lawyer have created a real business exploiting this theme” appears inconvenient. Authors should edit.

Lines 53-59: a proper reference is missing. This should be provided and moved in the section “discussion”.

Lines 68-71: this should be moved to the section “Discussion”

Lines 72-73: this is a personal opinion, it should be erased.

Lines 93-104: the English form must be edited.

Lines 104: since the same Institution has published a more recent paper involving the study period of the present manuscript, the authors should properly integrate this part with the following reference –

Treglia, M., Pallocci, M., Passalacqua, P., Giammatteo, J., De Luca, L., Mauriello, S., Cisterna, A. M., & Marsella, L. T. (2021). Medical Liability: Review of a Whole Year of Judgments of the Civil Court of Rome. International journal of environmental research and public health, 18(11), 6019. https://doi.org/10.3390/ijerph18116019

Point 3: Matherials and methods should include the type of statistical analysis or test used.

Point 4: RESULT SECTION

In figure 1 and figure 2, years are written in decimal form (e.g., 2005,00) and in the upper corner a mean and std dev of them are reported. This does not make sense and must be fixed.
All the following figures must be translated or corrected in English, as indicated in Point 1.

Line 228: You should provide the total number of the investigated professionals.

Figure 7 should report the same percentages of the lines 376-381

Point 5: the discussion should be implemented according to the issues of Point 2.

Moreover, the Authors should provide a proper reference for the following sections:

-          Lines 417-420;

-          Lines 444 – 447;

-          Lines 448-458

Point 6: Conclusions

Lines 460-461: a proper reference is missing and this should be provided and moved in the section “discussion”.

Lines 476-481: a proper reference is missing. This part should be provided and moved in the section “discussion”.

Avoid expressions like “ more relaxed manners” (lines 487-488) or “abolition of defensive medicine” (line 489.

Author Response

The language has been revised throughout the whole manuscript. However, as already mentioned it should be noted that some difficulties emerged when attempting to translate detailed notions of the Italian legal landscape into English.

For what concerns this point, We intentionally did not add to the legend the explanation to every single Article because in our opinion it would have make it too long and graphically less enjoyable, but the correspondence between every Article and the respective crime is easily available in the text.

Moreover, we apologize for the two graphs in Italian, it was certainly due to a copy-paste mistake during the layout phase.

Regarding the following points, we applied the requested changings.

Reviewer 4 Report

The article in question addresses a very pertinent and current problem. It gives a very valid contribution to the problem of the increase of litigation in the medical field and the so-called defensive medicine, since through a systematic collection of empirical elements on orthopedics malpractice claims in criminal court in the Italian context it allows to characterize this reality in a concrete space of time (2000-2015), while drawing conclusions about the parallels that this reality establishes with more transversal trends in this field. The analysis that is presented is clear and sufficiently detailed.

As regards some critical remarks that may contribute to reinforce the potential of this article, the following aspects are highlighted:

From the point of view of the contextualization of the phenomenon that is discussed, the introduction does not elaborate sufficiently on the multiple explanatory reasons that contribute to the expansion of this reality (it is exhausted in a single bibliographic reference) and gives a rather synthetic overview regarding the expression of this reality in the more general international context. Also on the issues of medical error and malpractice, it would be important to deepen and sophisticate the framing of these concepts, since it is a more complex reality than the premises presented give us to understand. The literature in this field is vast and although the objective is not to make this systematic review, it is still important to use some relatively unavoidable references that may contribute to the improvement of this discussion. The following stand out:

Merry, A.; McCall Smith, A. Errors, Medicine and the Law. Cambridge University Press: Cambridge, 2001.

Kohn, L.; Corrigan, J.; Donaldson, M., Eds. To Err is Human. Building a Safer Health System, National Academies Press: Washington (DC), 2000.

Harpwood, V. Medicine, Malpractice and Misapprehensions. Routledge-Cavendish: Abindon, 2007.

Cochrane, A. L. Effectiveness and efficiency. Random Reflections on Health Services. The Nuffield Provincial Hospitals Trust: Nuffield, 1972.

With regard to the discussion of the results, it would be very important to proceed with a more substantive analysis regarding the explanatory reasons that may shed light on the time trends presented. The analysis is excessively descriptive and not very analytical. It would also be important to further develop the discussion based on the conclusions obtained in order to highlight to what extent the trends found converge or diverge with the different configurations of the phenomenon in international terms. The argument that there are cross-cutting problems and trends is understandable, however there are quite distinct particularities between countries, especially when considering the reality of Europe and the United States. As such, it would not be irrelevant to elaborate on the differentiated configurations of this reality according to countries and health systems, given that the European model (although with particularities) is very different from the North American model.

It would also be important to develop a little more on the issue of the bureaucratisation of clinical practice. This is a very important critical point and deserves more emphasis.

In the conclusions (2nd paragraph, line 467), it would be important to indicate which are the most representative studies of the international literature referred to.

Finally, in formal terms, there are only a few lapses in the captions of figures 4 and 5. They appear in Italian, when they should be in English.

Author Response

We sincerely thank this Reviewer for highlighting the relevance of this topic and of the data our Manuscript presents.

As pointed out by the Reviewer, our aim was not to create a systematic review of this complex topic, but to enrich the available evidence with some insights that can be useful to make comparisons on a National and International level. We appreciate the Reviewer’s contribution and we have add some of the suggested quotes to the Introduction, to give a more detailed background to the matter.

We also deepened and implemented the Discussion and Conclusions as requested.

We apologize for the mistakes in English grammar and vocabulary, being able to attribute some of it to the difficulty of translating detailed notions of the Italian legal landscape into a foreign language.

Finally, we apologize for the two graphs in Italian, it was certainly due to a copy-paste mistake during the layout phase.
